# Methods

# A novel human organoid model system reveals requirement of TCF4 for oligodendroglial differentiation

Federica Furlanetto[1,*], Nicole Flegel[1,*], Marco Kremp[1,*], Chiara Spear[1], Franziska Fröb[1], Margherita Alfonsetti[1], Bettina Bohl[1], Mandy Krumbiegel[2], Sören Turan[1], Andre Reis[2], Dieter C Lie[1,3], Jürgen Winkler[4], Sven Falk[1], Michael Wegner[1], Marisa Karow[1]

**Heterozygous mutations of *TCF4* in humans cause Pitt–Hopkins syndrome, a neurodevelopmental disease associated with intellectual disability and brain malformations. Although most studies focus on the role of TCF4 in neural stem cells and neurons, we here set out to assess the implication of TCF4 for oligodendroglial differentiation. We discovered that both monoallelic and biallelic mutations in *TCF4* result in a diminished capacity to differentiate human neural progenitor cells toward myelinating oligodendrocytes through the forced expression of the transcription factors SOX10, OLIG2, and NKX6.2. Using this experimental strategy, we established a novel organoid model, which generates oligodendroglial cells within a human neurogenic tissue–like context. Also, here we found a reduced ability of TCF4 heterozygous cells to differentiate toward oligodendroglial cells. In sum, we establish a role of human TCF4 in oligodendrocyte differentiation and provide a model system, which allows to dissect the disease etiology in a human tissue–like context.**

## Introduction

Transcription factor 4 (*TCF4*), a member of the basic helix–loop–helix family (bHLH) of proteins, has been causally linked to intellectual disability and neuropsychiatric diseases such as schizophrenia, major depression, or autism spectrum disorder (Betancur, 2011; Vissers et al, 2016; Willsey et al, 2018). *TCF4* encodes a nuclear protein present not only in neurons, but also in substantial amounts in glial cells, including oligodendrocytes (OLs) of the central nervous system (Forrest et al, 2014). To perform its function, the class I bHLH protein TCF4 heterodimerizes with bHLH proteins from other classes (Forrest et al, 2014). Inactivating *TCF4* mutations

has been identified as the cause of the neurodevelopmental disorder Pitt–Hopkins syndrome (PHS; #610954; OMIM) whose symptoms include intellectual disability, failure to acquire language, deficits in motor learning, distinctive facial features, hyperventilation, gastrointestinal abnormalities, and autistic behavior (Zweier et al, 2007; Marangi & Zollino, 2015). In mouse models, only homozygous knockout of the *Tcf4* gene unfolds the whole spectrum of symptoms, whereas heterozygous mutations only result in a mild phenotype (Kennedy et al, 2016; Thaxton et al, 2018; Mesman et al, 2020; Wang et al, 2020). In contrast, in humans only heterozygosity of *TCF4* mutations is clinically relevant (Teixeira et al, 2021). Current studies on the role of human TCF4 during brain development and PHS are limited to the impact of *TCF4* mutations on neural progenitor cells (NPCs) and neurons. *TCF4* haploinsufficiency was, for example, found to play a role in synaptic development and plasticity of human cortical neurons (Davis et al, 2024), and in the proliferation of neural progenitors within brain organoids (Papes et al, 2022). There is, however, also compelling evidence for the role of Tcf4 in exerting relevant developmental functions within oligodendroglial lineages (Phan et al, 2020). As shown in our own previous work, Tcf4 is required for the differentiation of oligodendroglial progenitor cells into OLs, where it heterodimerizes with Olig2 and functionally interacts with Sox10, two most prominent oligodendroglial master regulators (Wedel et al, 2020). The relevance of the oligodendroglial TCF4 functions in the context of PHS is strongly supported by a study showing that in a Tcf4 mouse model of PHS, functional recovery can be induced by administration of promyelinating drugs (Bohlen et al, 2023). This highlights the need to assess the role of TCF4 in human oligodendrogenesis.

The human brain is composed of a magnitude of different cell types, which act in concert to ultimately exert the complex function of human cognition. With the advent of human brain organoids derived from human induced pluripotent stem cells (hiPSCs), we now have the opportunity to model especially early

---

[1]Institute of Biochemistry, Friedrich-Alexander-Universität Erlangen-Nürnberg, Erlangen, Germany    [2]Institute of Human Genetics, Friedrich-Alexander-Universität Erlangen-Nürnberg, Erlangen, Germany    [3]Institute of Anatomy, Friedrich-Alexander-Universität Erlangen-Nürnberg, Erlangen, Germany    [4]Department of Molecular Neurology, University Hospital Erlangen, Friedrich-Alexander-Universität Erlangen-Nürnberg, Erlangen, Germany

Correspondence: sven.falk@fau.de; michael.wegner@fau.de; marisa.karow@fau.de
Bettina Bohl's present address is Department of Bioengineering, Imperial College London, London, UK
*Federica Furlanetto, Nicole Flegel, and Marco Kremp contributed equally to this work

aspects of human brain development (Kelley & Pasca, 2022; Zhao & Haddad, 2024). However, the production of glial cells, in particular astrocytes and oligodendroglial cells, is rather limited in these 3D self-aggregated structures. Recently, new protocols focused on overcoming this limitation using patterning molecules and specific growth factors to promote the generation of oligodendroglial cells within brain organoids (Madhavan et al, 2018; Marton et al, 2019). Current protocols require a long time of differentiation and lead to the overrepresentation of one cell type, that is, oligodendrocytes.

With the ultimate goal of studying the impact of the clinically relevant TCF4 haploinsufficiency in human oligodendroglial cells, we took advantage of a cellular platform that allows directed differentiation of hiPSCs into human OLs in a timely controlled manner through the forced combinatorial expression of SOX10, OLIG2, and NKX6.2 (referred to as SON) (Ehrlich et al, 2017). We used these cells to generate human brain organoids that contain OLs and assessed the consequences of TCF4 haploinsufficiency in a human tissue–like context.

# Results and Discussion

### Generation of TCF4-deficient cells carrying the SON cassette

To study the role of TCF4 in human oligodendrocytes, a hiPSC line was used that was genetically modified by introducing a doxycycline (Dox)-inducible expression cassette for SOX10, OLIG2, and NKX6.2 (SON) into the adeno-associated virus integration site 1 locus. The joint expression of these three proteins had previously been shown to direct the differentiation of hiPSCs and hiPSC-derived NPCs into oligodendrocytes (Ehrlich et al, 2017). The resulting SON15 hiPSC line further underwent CRISPR/Cas9-dependent genome editing of the TCF4 locus. Using a guide RNA targeting the second helix of the bHLH DNA-binding domain of TCF4 in the carboxy-terminal part of the protein (Fig 1A), we obtained a heterozygous clone (TCF4 +/−, referred to as TCF4-HET) with a 19-bp deletion on one allele, and a homozygous clone that carried the same 19-bp deletion on one allele and an 8-bp deletion on the other allele resulting in compound heterozygous clone (referred to as TCF4-HOM, Fig 1B). All deletions led to frameshifts and in case of translation would truncate the protein and destroy its dimerization and DNA-binding ability. Determination of TCF4 transcript levels by quantitative RT–PCR revealed a significant reduction in both clones with a residual 49% in the TCF4-HET clone and almost negligible levels in the TCF4-HOM clone (Fig 1C). Compared with WT SON15 cells, TCF4 protein levels were also significantly reduced in the TCF4-HET clone and virtually undetectable in the TCF4-HOM clone as judged by immunocytochemical stainings and their quantification (Fig 1D and E). We conclude from these observations that TCF4 mutations lead to a severe reduction in both transcript and protein levels. The resulting reduction of the TCF4 protein in the TCF4-HET clone and the absence of the TCF4 protein in the TCF4HOM clone did not interfere with the pluripotency of the clones as evidenced by an expression of OCT4 and NANOG that was comparable to the original SON15 line in the unmodified hiPSC state (Fig 1D).

### TCF4-deficient cells show impaired oligodendroglial maturation

To assess the ability of the clones to differentiate into oligodendrocytes, we first differentiated all lines to SOX2, NESTIN, and PAX6 triple-positive NPCs (Fig 2A). Next, we performed bulk RNA sequencing to evaluate whether heterozygous or homozygous loss of TCF4 would impact on basic properties of the NPCs. This assessment revealed that overall, there were no major differences concerning the expression levels of pluripotency, proliferation, and neuronal, astroglial, or oligodendroglial markers across lines (Fig 2B). Concomitant with a gradual reduction of the TCF4 mRNA expression levels from TCF4-WT to TCF4-HET and even more to TCF4-HOM samples, the TCF4 regulon activity simultaneously decreased (Fig 2C and D) directly correlating with TCF4 mRNA expression (Fig 2E).

For consecutive differentiation into oligodendrocytes, NPCs were treated with Dox to turn on the expression of SOX10, OLIG2, and NKX6.2 for 12 consecutive days, during which the medium was changed after 2 d from standard NPC medium containing SAG (smoothened agonist) to PDGF-A (platelet-derived growth factor A) and insulin-like growth factor-1 (IGF-1)–containing glial induction medium and after another 4 d to thyroid hormone–containing glial differentiation medium (Fig 2F). Upon glial induction, ~60–70% of all cells expressed SOX10 and OLIG2, independent of their genotype (Fig 2G). We first checked for proliferation of cells 4 d after glial induction. We found no significant differences in the fraction of SOX10 and Ki-67 double-positive cells or SOX10, O4, and Ki-67 triple-positive cells over all SOX10-positive cells between TCF4-WT, TCF4-HET, and the TCF4-HOM samples (Fig 2H and I). At the same time, the number of cleaved caspase-3–positive (i.e., apoptotic) cells was below 0.1% and comparable between all three lines at all times tested.

After 18 d of differentiation, we stained the resulting oligodendrocytes using antibodies against SOX10, O4, and MBP (Fig 3A), showing that in principle, cells can differentiate to oligodendrocytes independent of the TCF4 genotype. However, qRT–PCR assessing the expression of myelin markers such as MBP, PLP1, MOG, CNP, and GALC revealed down-regulation of oligodendroglial lineage–associated genes in a TCF4 genotype–dependent manner. Furthermore, transcripts related to lipid metabolism such as ACSS2, FASN, and HMGCR were substantially down-regulated in clones carrying either heterozygous or homozygous TCF4 mutations. Of note, the pre-myelinating marker NKX2.2 did not change (Fig 3B).

Next, we performed a quantification of O4- and MBP-positive cells (Fig 3C). The O4 antibody recognizes a sulfated surface galactocerebroside that appears during oligodendrocyte development at early stages of differentiation in the so-called pre-myelinating oligodendrocytes (Sommer & Schachner, 1982). Quantification of the fraction of O4-positive cells revealed a significant decrease in the TCF4-HOM samples, indicating that TCF4 is required for proper differentiation to this stage. Staining for the myelin component MBP expressed later during oligodendrocyte development pointed to a significant reduction, indicating that the conversion of pre-myelinating oligodendrocytes into myelinating oligodendrocytes is strongly impaired in both TCF4-HET and TCF4-HOM cells (Fig 3A and C). Similarly, the fraction of MBP-positive cells among the O4-positive cells was substantially reduced in both TCF4-HET and TCF4-

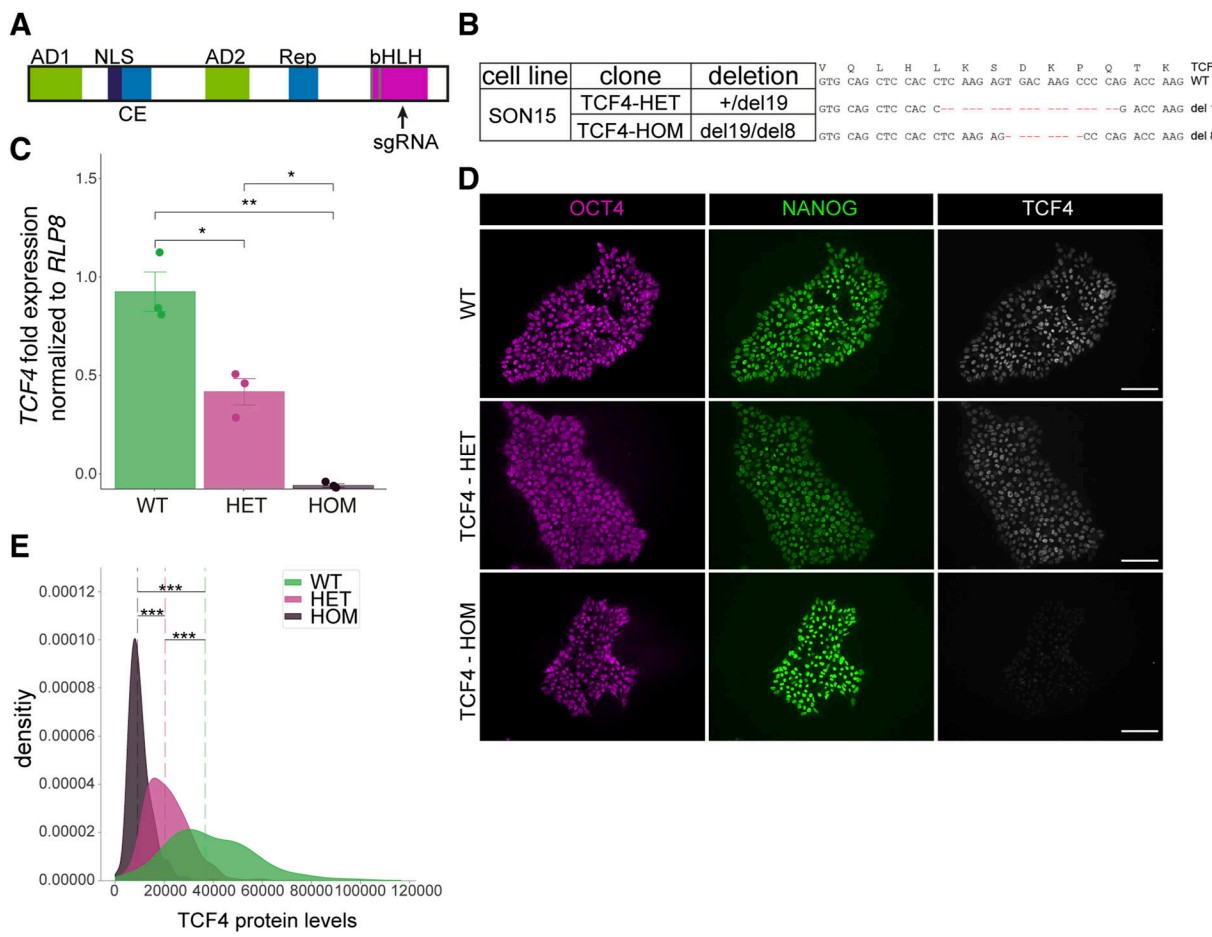

**Figure 1. Generation of TCF4-deficient hiPSCs carrying the SON cassette.**
**(A)** Scheme depicting the TCF4 protein including the main functional domains (AD, activation domain; NLS, nuclear localization signal; CE, repressor domain; Rep, repressor domain; bHLH, DNA-binding domain) and the localization of the guide RNA (sgRNA) targeting the bHLH domain of TCF4. **(B)** Summary of the SON15 clones used in the study and the CRISPR/Cas9-induced deletions (del) present in them. **(C)** Quantitative RT–PCR of *TCF4* transcripts in the TCF4-HET and TCF4-HOM clones relative to the original SON15 cell line (set to 1) and normalized to the housekeeping gene *RLP8*. Shown are mean values ± SEM (*n* = 3). **$P < 0.005$, *$P < 0.05$, *t* test. Exact *P*-values (top to down): 0.018, 0.0098, 0.018. **(D)** Immunofluorescence stainings of SON15 hiPSCs and genome-edited clones confirming pluripotency of the lines through OCT4 and NANOG expression and simultaneous assessment of TCF4 expression. Note the gradually diminished expression of TCF4 in the HET and the HOM clones. Scale bars = 100 μm. **(E)** Quantification of TCF4-specific immunocytochemical signal in SON15 hiPSCs and their TCF4 variant clones. Dotted lines represent the median. Exact *P*-values (top to down): $3.4 \times 10^{-182}$, $4.6 \times 10^{-193}$, $1.3 \times 10^{-196}$; ***$P < 0.001$. *n* = 542 cells, WT; *n* = 1,159 cells, HET; *n* = 842 cells, HOM; Mann–Whitney *U* test.

HOM clones (Fig 3D). Furthermore, quantification of the intensities of the MBP signals across samples unraveled that not only the number of the MBP-expressing cells decreased in TCF4-deficient cells, but within the MBP-positive cells, the amount of the myelin proteins was reduced upon TCF4 down-regulation (Fig 3E). Besides the number of O4 cells, we also assessed the morphology of the cells, which acquired O4 immunoreactivity. We categorized the cells as exhibiting either a more "compact" or more "complex" elaborated morphology (see examples in Fig 3F). Here, in both conditions we found an increase in compact cells and a concomitant decrease in complex cells dependent on the TCF4 genotype at two different timepoints of differentiation (Fig 3G). A comparable differentiation defect has previously been detected in Tcf4-deficient mouse oligodendrocytes in vivo and in vitro (Wedel et al, 2020). However, heterozygous loss of TCF4 did not impact OL differentiation in the mouse, indicating that the TCF4 threshold levels required for proper myelination may be higher in the human system.

**Generation of oligodendroglial cells within brain organoids**

Based on the new finding that in contrast to the mouse, already heterozygous *TCF4* loss leads to differentiation deficits in human oligodendroglial cells, we set out to assess the impact of *TCF4* heterozygosity specifically in the oligodendroglial lineage in a human neurogenic tissue–like context provided by brain organoids. To overcome the limitation that unguided brain organoids usually generate few oligodendroglial cells, we employed a mixing approach in which we jointly aggregated control hiPSCs with SON15 hiPSCs in a ratio of 8:1 (Fig 4A). To induce the SON cassette, we treated mixed organoids with Dox starting on day 30 after aggregation of the hiPSCs and added PDGF-A and IGF-1 to the medium to support oligodendroglial induction. From day 40 onward, we in addition added T3 (triiodothyronine) along with Dox to foster oligodendroglial differentiation. Although OLIG2-expressing cells were found intermingled with other organoid-resident cells after

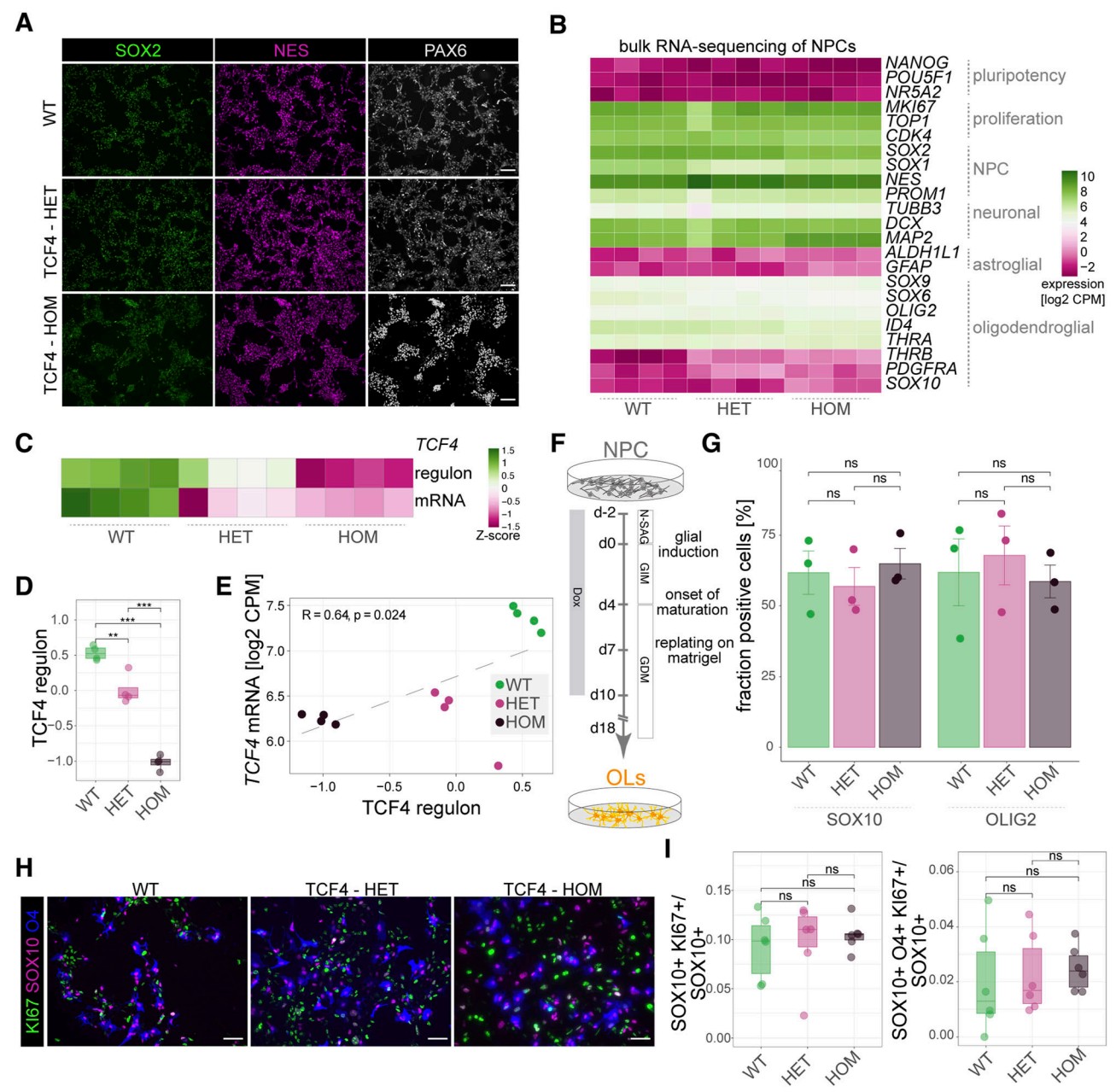

**Figure 2. Characterization of TCF4 variant cells.**
**(A)** Immunocytochemical stainings of NPCs derived from SON15 hiPSCs and genome-edited clones for SOX2 (green), NESTIN (NES; magenta), and PAX6 (gray). Scale bars = 100 μm. **(B)** Heatmap showing the expression (log₂ of counts per million) of selected pluripotency, proliferation, NPC, and neuronal, astroglial, and oligodendroglial markers across samples. There are four replicates per condition. **(C)** Heatmap showing the Z-score of TCF4 regulon activity (upper row) and *TCF4* mRNA expression (lower row) across samples. **(D)** Quantification of TCF4 regulon activity across samples as shown by box plot and jitters. *n* = 4 for all conditions; exact *P*-values (top to bottom): 0.0000121, 0.0000004, 0.001958. Boxplots show median, quartiles (box), and range (whiskers). **(E)** Dot plot indicating the correlation between *TCF4* mRNA expression and TCF4 regulon activity. *n* = 4 for all conditions; Pearson's correlation coefficient and *P*-value are indicated inside the plot. **(F)** Timeline showing the different steps of the oligodendroglial differentiation protocol from the NPC stage (N-SAG, neural progenitor medium containing SAG [smoothened agonist]; GIM, glial induction medium; GDM, glial differentiation medium). **(G)** Quantification of the fraction of SOX10- and OLIG2-expressing NPCs upon Dox induction across experimental lines shows no difference. *n* = 3. **(H)** Images showing NPCs 4 d after glial induction and stained for the proliferation marker Ki-67 (green), SOX10 (magenta), and O4 (blue). Scale bars = 100 μm. **(I)** Quantification of the fraction of double (left panel, SOX10+ Ki-67+)- and triple-positive (right panel, SOX10+ O4+ Ki-67+) cells over SOX10-positive cells as indicated in the graphs. Left and right plot: WT: *n* = 6; HET: *n* = 6; HOM: *n* = 6. Boxplots show median, quartiles (box), and range (whiskers). For (D, E, G, I), one-way ANOVA with Tukey's post hoc test. **<0.01, ***P < 0.001, ns = nonsignificant.

Dox treatment (Fig 4B), we did not find OLIG2-expressing cells in mixed organoids without Dox treatment in line with the commonly observed absence of oligodendroglial cells in unguided brain organoids (Fig 4C). After Dox treatment for at least 10 d, we observed the co-expression of OLIG2 with CNP (Fig 4D and E), a marker for early oligodendrocytes (Lappe-Siefke et al, 2003). MBP expression in

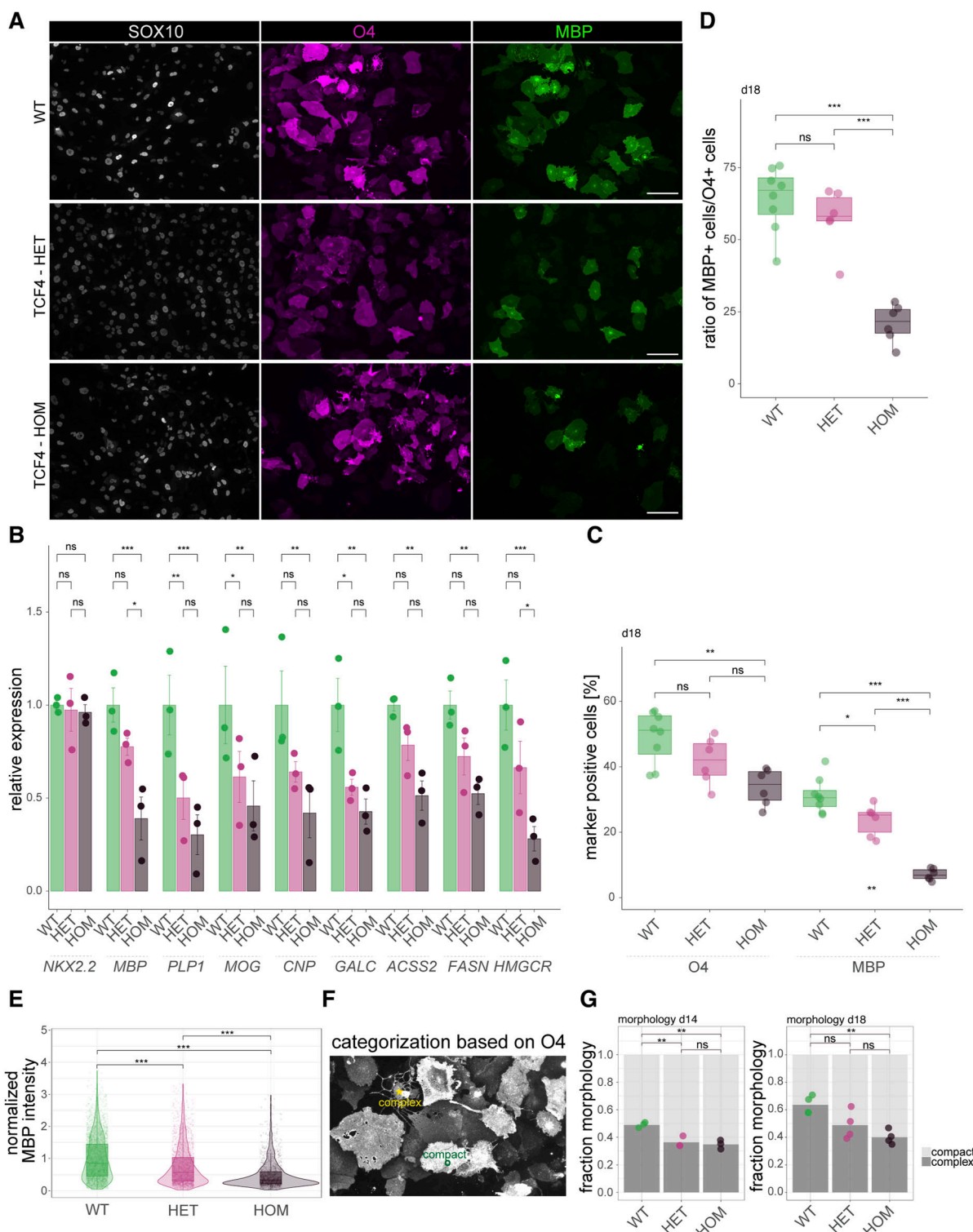

**Figure 3. Oligodendroglial differentiation deficit in TCF4 variant cells.**
**(A)** Representative images showing stainings using antibodies against SOX10 (gray), O4 (magenta), and MBP (green) to visualize the efficiencies of oligodendroglial differentiation in SON15 hiPSC-derived cells with and without *TCF4* mutations at day 18 after onset of differentiation. Scale bars = 100 μm. **(B)** Quantitative RT–PCR of transcripts for the pre-myelinating marker *NKX2.2*, myelin proteins (*MBP, PLP1, MOG, CNP, GALC*), and lipid metabolic enzymes (*FASN, HMGCR*) in the TCF4-HET and TCF4-HOM cells relative to the original SON15 cells (set to 1; dotted line) after 18 d of differentiation. Shown are mean values ± SEM (*n* = 3). Exact *P*-values (left to right, top to bottom): 0.0007, 0.01493, 0.0001, 0.0062, 0.0028, 0.0421, 0.0013, 0.0015, 0.0171, 0.0078, 0.0093, <0.0001, 0.044. *P < 0.05, **P < 0.01, ***P < 0.001, ns = nonsignificant. Two-way ANOVA with Tukey´s post hoc test. **(C)** Quantification of O4- and MBP-positive myelinating oligodendrocytes after 18 d of differentiation normalized to all SOX10-positive cells. Exact *P*-values (top to bottom): O4: 0.0027, 0.1580, 0.1713; WT: *n* = 7, TCF4-HET: *n* = 5, TCF4-HOM: *n* = 5; MBP: 3.8 × 10⁻⁸, 1.6 × 10⁻⁵, 0.0142; ***P < 0.001, **P < 0.01, *P < 0.05.

the OLIG2-positive SON15 cells was detectable in the mixed organoids after Dox treatment for 20 d, but not 10 d (Fig 4F and G). To characterize the SON cassette–containing cells within the human tissue–like niche of brain organoids further, we determined the fraction of OLIG2-positive cells co-expressing Ki-67, a marker for proliferating cells. This revealed a rather small fraction of OLIG2-expressing cells which also proliferated (Fig 4H and I).

This mixed organoid model allows to address an intriguing aspect of how a change in the cellular composition of brain organoids would feedback on the overall composition and development of brain organoid–resident cells. To unambiguously trace the oligodendroglial SON cells, we set out to fluorescently label the SON15 hiPSC line (Fig 4J). Next, we used this GFP-labeled SON15 line for the generation of mixed organoids with a WT ctrl hiPSC line (Fig 4K). This approach allowed us to determine the induction efficiency upon Dox treatment (Fig 4L) and revealed a strikingly lower induction efficiency in the 3D (Fig 4L) versus the 2D setting, where a fraction of about 60% of the cells showed the expression of the *SOX10*, *OLIG2*, and *NKX6.2* transgenes (see Fig 3B). Interestingly, we sometimes also detected OLIG2-expressing cells without a GFP label in addition to cells both positive for OLIG2 and GFP. In future experiments, it will be interesting to study a putative induction of oligodendroglial differentiation in non-SON cells by the presence of SON-derived oligodendroglial cells. Furthermore, future experiments will have to address whether SON-derived OLs are able to myelinate organoid-resident neurons as proposed in an earlier study where induction of oligodendroglial cells in organoids resulted in the presence of myelin (Madhavan et al, 2018; Ng et al, 2021).

### Assessment of oligodendroglial cells with heterozygous TCF4 loss in brain organoids highlights the potential of using mixed organoids for studying disease etiology

Finally, we set out to use our mixed organoids to study the impact of heterozygous *TCF4* loss in a 3D human tissue–like context. Because human patients with PHS are heterozygous carriers of *TCF4* mutations (Zweier et al, 2007), we used the TCF4-HET SON15 line (Fig 1B) for mixed organoid generation (Fig 5A). We followed the same scheme and treated the mixed organoids for 10–20 d with Dox to induce the SON cassette and promote oligodendroglial differentiation. We first assessed whether there is a difference in the fraction of proliferating Ki-67 and OLIG2 double-positive cells (Fig 5B) between the TCF4-WT and the TCF4-HET condition. However, no significant alteration in the proliferation capability of the SON cells within brain organoids was observed (Fig 5C). As a readout for differentiation, we quantified the fraction of OLIG2/CNP-co-expressing cells over all OLIG2-expressing cells in both conditions and found a significant reduction of the differentiated cells in

organoids with TCF4-HET cells (Fig 5D and E). We next quantified the fraction of OLIG2/MBP-co-expressing cells over all OLIG2-positive cells within brain organoids (Fig 5F) and found a strong reduction of MBP-expressing SON15 cells (OLIG2 positive) in the TCF4-HET condition. This finding further corroborates the 2D findings that in human oligodendrocyte development, the reduction of TCF4 dosage in heterozygous TCF4 mutants is sufficient to elicit differentiation deficits. To exclude that cell death of TCF4-deficient oligodendroglial cells is the underlying reason for the differentiation deficit, we determined whether apoptosis is changed in TCF4-HET cells and stained for PARP1 (Fig 5G). Analysis of these stainings showed that almost no OLIG2 cells underwent apoptosis whether in TCF4-WT nor TCF4-HET conditions (Fig 5H). In sum, our data provide compelling evidence that unlike in mice, in humans heterozygous TCF4 ablation results in oligodendrocyte differentiation deficits also in a tissue-like context.

Importantly, by mixing genetically modified hiPSCs carrying the SON cassette with WT cells, we will be able to assess the impact of the manipulated oligodendroglial lineage cells on WT NPCs and neurons. This will allow to deconstruct the consequences of a mutation and the series of events leading to the development of patient phenotypes.

In sum, we not only provide strong evidence for the translation of earlier findings on the role of TCF4 in murine oligodendrogenesis to human OL production but furthermore introduce a novel model system to assess cellular interactions between human oligodendroglial and neural cells in a human tissue–like context.

## Materials and Methods

### Culturing and differentiation of hiPSCs

For the current study, a hiPSC line (UKERiO3H-S1-006) was used that was generated from dermal fibroblasts of a 71-yr-old healthy male and genetically modified by integration of a Dox-inducible expression cassette for *SOX10*, *OLIG2*, and *NKX6.2* (SON cassette) into the adeno-associated virus integration site 1. This cell line is referred to as SON15. Karyotyping revealed a balanced translocation between the Y chromosome and one chromosome 20. In most cells, an additional derivative chromosome 20 was present. In a small percentage of cells, an isochromosome 20q was detectable instead of the two derivative chromosomes 20. This kind of chromosomal alterations is frequent in pluripotent stem cells (Avery et al, 2013). SON15 hiPSCs were cultured on 6-well plates coated with Matrigel growth factor reduced (Thermo Fisher Scientific) in mTeSR Plus medium (StemCell Technologies) with penicillin–streptomycin (Anprotec). Medium change was performed daily, passaging once a

---

WT: $n$ = 7, TCF4-HET: $n$ = 5, TCF4-HOM: $n$ = 5. Boxplots show median, quartiles (box), and range (whiskers). **(D)** Box-and-jitter plot showing the fraction of MBP-positive cells over O4 cells. Exact $P$-values (top to down): 0.0000010, 0.0000206, 0.4403782; WT: $n$ = 8, TCF4-HET: $n$ = 6, TCF4-HOM: $n$ = 6; boxplots show median, quartiles (box), and range (whiskers). **(E)** Quantification of normalized MBP intensity across conditions as shown by box, violin, and jitter plot. Exact $P$-values for all comparisons: $<2.2 \times 10^{-308}$; WT: $n$ = 3,158, TCF4-HET: $n$ = 2,135, TCF4-HOM: $n$ = 1,404. Boxplots show median, quartiles (box), and range (whiskers). **(F)** Example of O4-positive cells with complex (circle) and compact (star) morphology. **(G)** Quantification of the fraction of compact and complex cells across conditions at d14 (left panel) and d18 (right panel) upon induction of oligodendroglial differentiation. Exact $P$-values (top to bottom): d14: 0.0039, 0.0068, 0.8347; d18: 0.0051, 0.0582, 0.2960; $n$ = 4 for all conditions. For (C, D, E, G), $*P < 0.05$, $**P < 0.01$, $***P < 0.001$, ns = nonsignificant. One-way ANOVA with Tukey's post hoc test.

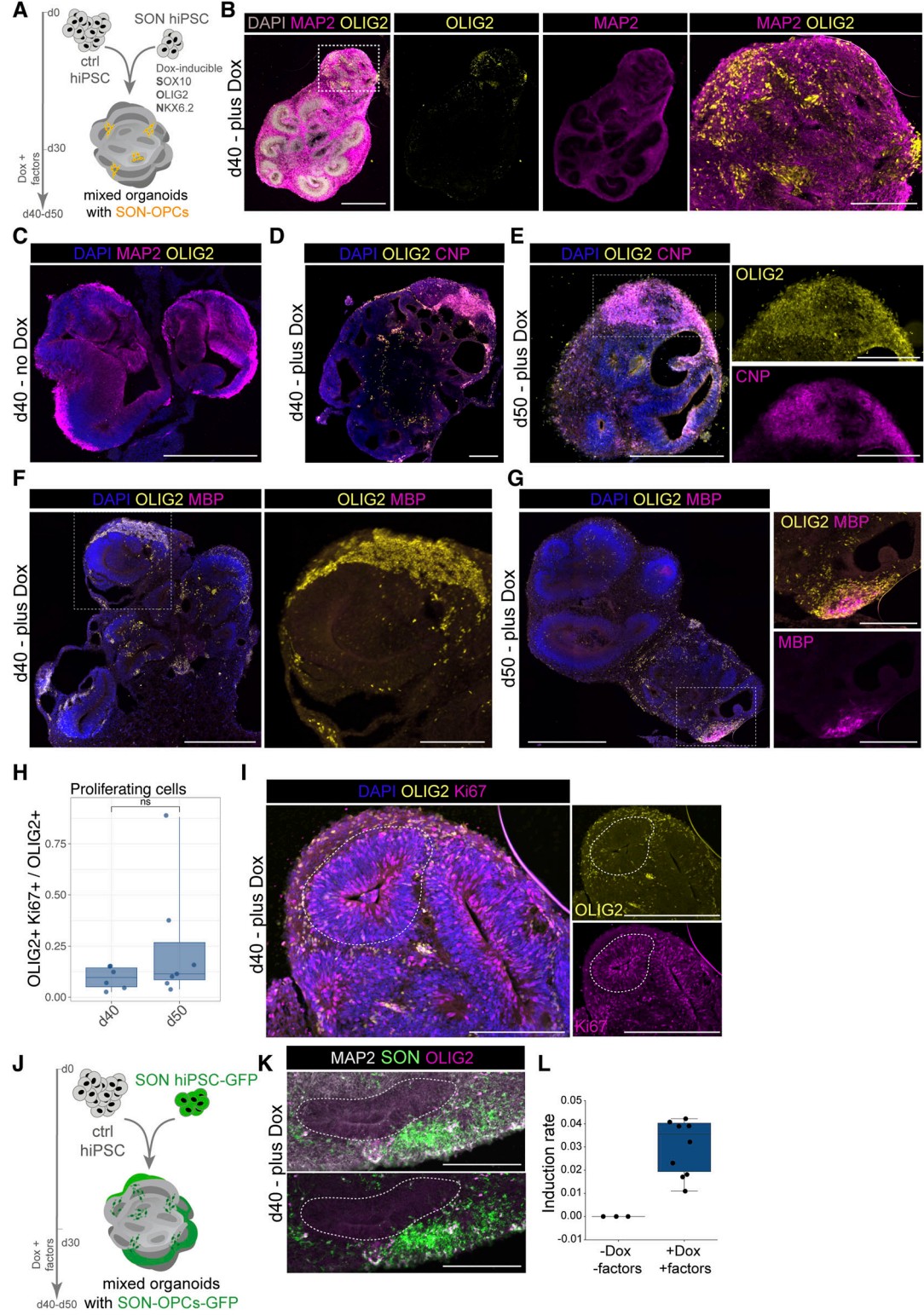

**Figure 4. Mixed organoids contain oligodendroglial cells.**
**(A)** Experimental scheme for generating mixed organoids using control (ctrl) hiPSC and SON cassette containing hiPSCs. Dox induction starting on d30 after aggregation of hiPSCs induces *SOX10*, *OLIG2*, and *NKX6.2* expression in organoid-resident SON cells. **(B)** Immunofluorescence stainings against MAP2 (magenta) and OLIG2 (yellow) to visualize cellular organization and cell types in organoid slices on d40 post-initial cell aggregation, that is, 10 d of Dox treatment. The right panel shows zoom-in as delineated in the overview picture on the left. DAPI (gray) was used to counterstain nuclei. Scale bars = 500 μm; 200 μm for zoom-in. **(C)** Organoid slice of a mixed organoid without Dox treatment stained for MAP2 (magenta) and OLIG2 (yellow) confirms the absence of OLIG2-expressing cells in these organoids. Scale bar = 500 μm. **(D)**

week using ReLeSR (StemCell Technologies) according to the manufacturer's instructions.

For the generation of hiPSC-derived NPCs, hiPSCs were cultured on a layer of mitomycin C–treated primary mouse embryonic fibroblasts in hES medium, consisting of DMEM/F12 + GlutaMAX (Thermo Fisher Scientific), 20% KnockOut Serum Replacement (Thermo Fisher Scientific), FGF2 (PeproTech), 1% penicillin–streptomycin, 1% nonessential amino acids (Thermo Fisher Scientific), and 0.05 mM 2-mercaptoethanol (Roth). After reaching 80% confluency, hiPSCs were detached with ReLeSR and expanded on Matrigel-coated dishes. For embryoid body (EB) formation, cells were transferred in ultra-low attachment dishes (Corning) and cultured in hES medium containing dorsomorphin (1 $\mu$M; Cell Guidance Systems), Chir99021 (3 $\mu$M; Cell Guidance Systems), and SB431542 (10 $\mu$M; PeproTech). On the second day, medium was changed to N2B27 medium composed of Neurobasal medium (Thermo Fisher Scientific), DMEM/F12 + GlutaMAX, 0.5 × N2 and 0.5 × B27 supplement w/o vitamin A (Thermo Fisher Scientific), dorsomorphin (1 $\mu$M), Chir99021 (3 $\mu$M), purmorphamine (0.5 $\mu$M; Tocris Bioscience), and SB431542 (10 $\mu$M). On day 4, dorsomorphin was replaced by ascorbic acid (150 mM; Sigma-Aldrich). On day 6 of differentiation, around 20–30 EB colonies were picked and transferred to a Matrigel-coated plate containing the same medium. EB colonies were dissociated and resulting cells plated. From passages 0–4, NPCs were split at a ratio of 1:2–1:6, and afterward at a higher ratio of 1:8–1:10. From passage 6 on, the ROCK inhibitor Y-27632 (RI, 2 $\mu$M; Selleck Chemicals) was added and purmorphamine was replaced by smoothened agonist (SAG, 0.5 $\mu$M; Cayman Chemicals) resulting in a medium referred to as N-SAG.

For oligodendrocyte differentiation, NPCs were used from passage 6 onward. Approximately 1 × 10$^5$ cells per well of a 12-well plate were seeded, and doxycycline (3 $\mu$g/ml; Sigma-Aldrich) was added to the medium for the induction of the SON cassette (day −2 of oligodendrocyte differentiation). On day 0 of differentiation, the medium was changed to glial induction medium consisting of DMEM/F12 + GlutaMAX, 1% penicillin–streptomycin, 0.5 × N2 and 0.5 × B27 supplement w/o vitamin A, 0.5 $\mu$M SAG, 10 ng/ml PDGF-AA (PeproTech), 10 ng/ml NT-3 (PeproTech), 10 ng/ml IGF-1 (R&D Systems), 10 ng/ml T3 (Sigma-Aldrich), 200 $\mu$M ascorbic acid, 0.1% Trace Element B (Corning), and 3 $\mu$g/ml doxycycline. On day 4 of differentiation, medium was changed to glial differentiation medium composed of DMEM/F12 + GlutaMAX, 1% penicillin–streptomycin, 0.5 × N2 supplement, 0.5 × B27 supplement w/o vitamin A, 10 ng/ml NT-3, 10 ng/ml IGF-1, 10 ng/ml T3, 200 $\mu$M ascorbic acid, 0.1% Trace

Element B, 100 $\mu$M N6, 2'-O-dibutyryladenosine 3': 5'-cyclic monophosphate sodium salt (dbcAMP; Sigma-Aldrich), and 3 $\mu$g/ml doxycycline. On day 7, cells were replated at a density of 1 × 10$^5$ cells per well of a 12-well plate, or at a density of 0.5 × 10$^5$ cells per well of a 24-well plate. On day 10, Dox was withdrawn. By day 18, the cells were terminally differentiated and analyzed by flow cytometry or immunocytochemistry (see below). All cells were regularly tested for the presence of *Mycoplasma* using PCR *Mycoplasma* Test Kit (PromoKine) or LookOut *Mycoplasma* PCR Detection Kit (Sigma-Aldrich).

## Genome editing of hiPSCs

For genome editing, the plasmid pCAG-SpCas9-GFP-U6-gRNA (Addgene) was used that expressed the 5'-CCACCTCAAGAGTGA-CAAGCCCC-3' guide RNA under the control of the CAG promoter in addition to Cas9 and GFP. hiPSCs were nucleofected using Amaxa P3 Primary Cell 4D-Nucleofector X Kit S (Lonza). 2 d after nucleofection, the cells were detached with Accutase (Thermo Fisher Scientific) and transferred to a falcon tube. GFP-expressing cells underwent FACS and were cultivated as single cells per well in a 96-well plate on Matrigel in mTeSR medium with 10% CloneR (StemCell Technologies). Arising colonies were expanded for freezing of stocks and analysis. Resulting hiPSC clones were genotyped by PCR amplification of the *TCF4* gene using 5'-TGCCTGCTTTGCAGAGTGTA-3' and 5'-GGTGAACTGCATGTGAGTGTG-3' as primers, and Sanger sequencing of the PCR product, and consecutive analysis was conducted using the tool on the ICE Synthego website (https://ice.synthego.com/). In a similar manner, the status of the most likely off-target genes was checked and found to be unaltered. Karyotyping of the TCF4-HET and the TCF4-HOM lines revealed no additional alterations other than the ones detected in TCF4-WT cells.

## Generation of stable GFP-expressing SON-hiPSCs

For the generation of the stable GFP-expressing hiPSC line, we used piggyBac-mediated insertion (Chen & LoTurco, 2012) of the fluorescent reporter GFP under the control of a ubiquitin C promoter. pUB-GFP (Matsuda & Cepko, 2004) was a gift from Connie Cepko (plasmid # 11155; Addgene). hiPSCs were electroporated using Human Stem Cell Nucleofector Kit 2 (Lonza). Approximately 2.2 million cells were detached with Accutase and pelleted per transfection reaction. Cells were resuspended in 100 $\mu$l of mixed supplement solution 1 and Nucleofector solution 2 (Human Stem Cell Nucleofector Kit 2) plus

Immunofluorescence staining of mixed organoid containing OLIG2 (yellow)- and CNP (magenta)-positive SON15 cells on d40 after aggregation of hiPSCs, that is, 20 d of Dox treatment. DAPI (blue) was used to counterstain nuclei. Scale bar = 200 $\mu$m. **(E)** Representative image showing the existence of OLIG2 (yellow)- and CNP (magenta)-positive SON15 cells on d50 after aggregation of hiPSCs, that is, 30 d of Dox treatment. DAPI (blue) was used to counterstain nuclei. Scale bar = 500 $\mu$m; 200 $\mu$m for zoom-ins. **(F)** Image showing mixed organoid slice (d40) containing OLIG2-expressing cells (yellow). Note that at this timepoint (10 d Dox treatment), no MBP expression can be detected. Zoom-in taken from overview in small inset (within a dashed line). Scale bar = 500 $\mu$m; 200 $\mu$m for zoom-in. **(G)** Image of an organoid slice confirms the appearance of OLIG2 (yellow)-expressing cells exhibiting oligodendroglial morphology as shown by MBP (magenta) expression on d50. Zoom-in taken from overview in small inset (within a dashed line). DAPI (blue) was used to counterstain nuclei. Scale bar = 500 $\mu$m; 200 $\mu$m for zoom-ins. **(H)** Quantification of the fraction of proliferating cells as determined by Ki-67 and OLIG2 staining. ns = nonsignificant. d40: $n$ = 6; d50: $n$ = 7; $n$ represents individual ventricles of three different organoids. Boxplots show median, quartiles (box), and range (whiskers). **(I)** Image showing examples of Ki-67- and OLIG2-positive cells in d40 organoid slices 10 d post-Dox treatment. DAPI (blue) was used to counterstain nuclei. Scale bar = 200 $\mu$m. **(J)** Graphical scheme describing the experimental setup to constitutively label SON cells. **(K, L)** Immunofluorescence stainings of mixed organoids containing GFP-expressing SON cells. Note the co-expression of OLIG2 (white) and GFP allowing quantification of the induction rate as shown in (L). Scale bar = 200 $\mu$m. **(L)** Induction rate of the SON cassette upon Dox treatment of mixed organoids as quantified by OLIG2-positive cells over GFP-positive SON15 hiPSCs. $n$ = 3 for -Dox, $n$ = 9 for +Dox; $n$ represents individual organoids. Boxplots show median, quartiles (box), and range (whiskers).

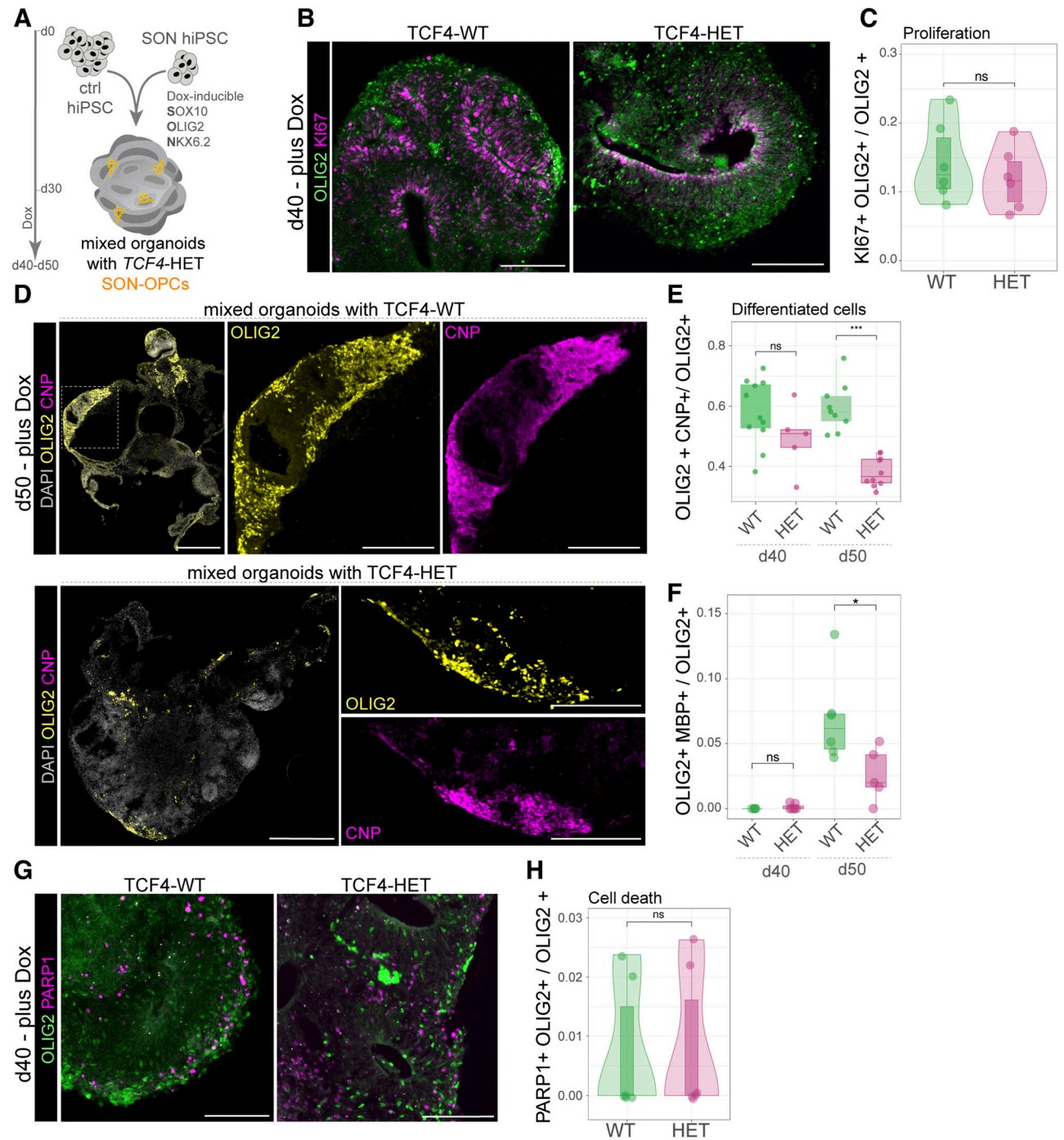

**Figure 5. Assessment of TCF4 deficiency in oligodendroglial cells reveals impaired differentiation within a human tissue–like context.**
**(A)** Experimental scheme for generating mixed organoids using control (ctrl) hiPSC and SON15 hiPSCs either expressing WT or being heterozygously mutant for *TCF4*. Dox induction follows the same timeline as described in Fig 4A. **(B)** Images showing OLIG2 (green) and Ki-67 (magenta) immunoreactivity in mixed organoids (d40). Scale bars = 500 μm. **(C)** Box, violin, and jitter plot showing the quantification of Ki-67 and OLIG2 double-positive cells over all OLIG2-positive cells. Exact *P*-value: 0.4432. WT: *n* = 6 organoids; HET: *n* = 6 organoids. **(D)** Pictures showing CNP-expressing cells exhibiting oligodendroglial morphology within brain organoids generated with TCF4-WT SON15 cells (upper panel) or TCF4-HET SON15 cells (lower panel). DAPI (gray) was used to counterstain nuclei. Scale bars = 500 μm; 200 μm for zoom-ins. **(E)** Quantification of the fraction of OLIG2 and CNP double-positive cells over total OLIG2-positive cells at d40 and d50. Exact *P*-values: d50: 1.6 × 10$^{-6}$, d40: 0.1645. WT: *n* = 11 organoid slices from 1 batch for d40, 9 organoid slices from 1 batch for d50; HET: *n* = 5 organoid slices from 1 batch for d40, 8 organoid slices from 1 batch for d50. **(F)** Quantification of the fraction of OLIG2 and MBP double-positive cells over total OLIG2-positive cells at d40 and d50. Exact *P*-values (top to bottom): 0.01485, 0.9996. WT: d40: *n* = 4 organoids, d50: *n* = 6 organoids; HET: d40: *n* = 7 organoids, d50: *n* = 5 organoids. **(G)** Pictures showing immunohistochemical stainings of OLIG2 (green) and the apoptosis marker PARP1 (magenta). Scale bars = 500 μm. **(H)** Quantification of PARP1 and OLIG2 double-positive cells over OLIG2-positive cells in mixed organoids (d40, d50) as shown by box, violin, and jitter plots. Exact *P*-value: 0.9245. WT: *n* = 6 organoids, HET: d40: *n* = 6 organoids. For (C, E, F, H), *<0.05, ***<0.001, ns = nonsignificant. One-way ANOVA with Tukey´s post hoc test. Boxplots show median, quartiles (box), and range (whiskers).

1.5 µg of the pUB-GFP and 1.5 µg of piggyBac transposase pBase plasmid. Cell–plasmid solution was transferred to the electroporation cuvette (Human Stem Cell Nucleofector Kit 2), electroporated using the program CB-150 of the Nucleofector, and then transferred to a new well of a Matrigel-coated 6-well plate with 2 ml of mTeSR1 (StemCell Technologies) supplemented with Y-27623 ROCK inhibitor (10 µM; StemCell Technologies). 8 d post-transfection, GFP+ cells were sorted using FACS and replated. GFP-positive colonies were then manually picked and cultured in mTeSR Plus1 medium.

## Formation of mixed organoids

mTeSR Plus medium (StemCell Technologies) was used to culture all hiPSC lines used in this study. Cells were grown on Matrigel-coated six-well plates in 5% $CO_2$ at 37°C until a confluency of 80–90% was reached. Brain organoid formation was achieved using a published protocol including small adaptations (Lancaster et al, 2013). Briefly, Accutase (Thermo Fisher Scientific) was used to generate single-cell suspensions of hiPSCs. After centrifugation, cells were resuspended in organoid formation medium (OFM) supplied with 4 ng/ml of low bFGF (PeproTech) and 5 µM ROCK inhibitor Y-27632 (StemCell Technologies). OFM consisted of DMEM/F12 + GlutaMAX-I (Thermo Fisher Scientific), 20% KOSR (Thermo Fisher Scientific), 3% FBS (Thermo Fisher Scientific), 0.1 mM MEM-NEAA (Thermo Fisher Scientific), and 0.1 mM 2-mercaptoethanol (Sigma-Aldrich). 9,000 cells in 150 µl OFM/well were aggregated in low attachment 96-well plates (Corning) for at least 48 h during which embryoid bodies (EBs) were formed. For mixed organoids, we aggregated 8,000 hiPSCs with 1,000 SON15 hiPSCs. After 72 h, half of the medium was replaced with 150 µl of new OFM without bFGF and ROCK inhibitor. At day 5, neural induction medium consisting of DMEM/F12 + GlutaMAX-I (Gibco), 1% N2 supplement (Gibco), 0.1 mM MEM-NEAA (Gibco), and 1 µg/ml heparin (Sigma-Aldrich) was added to the EBs in the 96-well plate to promote their growth and neural differentiation. Neural induction medium was changed every 2 d until day 12/13, when aggregates were transferred to undiluted Matrigel (Corning) droplets. The embedded organoids were transferred to a petri dish (Greiner Bio-One) containing organoid differentiation medium (ODM) without vitamin A. 3 or 4 d later, the medium was exchanged with ODM with vitamin A and the plates were transferred to an orbital shaker set to 30 rpm inside the incubator. Medium was changed twice per week. ODM consisted of a 1:1 mix of DMEM/F12 + GlutaMAX-I (Thermo Fisher Scientific) and Neurobasal medium (Thermo Fisher Scientific), 0.5% N2 supplement (Thermo Fisher Scientific), 0.1 mM MEM-NEAA (Thermo Fisher Scientific), 100 U/ml penicillin and 100 µg/ml streptomycin (Thermo Fisher Scientific), 1% B27 +/− vitamin A supplement (Thermo Fisher Scientific), 0.025% insulin (Sigma-Aldrich), and 0.035% 2-mercaptoethanol (Sigma-Aldrich).

## Organoid fixation

For fixation, organoids were transferred to 1.5-ml tubes. Organoids were washed with PBS and then fixed with 1xPBS-buffered 4% PFA (Carl Roth) for 30 min. The time of PFA fixation was extended up to 1 h depending on the size of the organoids. Afterward, organoids were washed three times for 10 min with PBS and incubated in 30% sucrose (Sigma-Aldrich) in PBS for cryoprotection. For cryosectioning, organoids were embedded in Neg-50 Frozen Section Medium (Thermo Fisher Scientific) on dry ice. Frozen organoids were cryosectioned in 30-µm sections using the Thermo Fisher CryoStar NX70 cryostat. Sections were placed on SuperFrost Plus Object Slides (Thermo Fisher Scientific) and stored at −20°C until use.

## SON cassette induction in organoids

Mixed organoids at day 30 were placed into a six-well plate containing ODM + vitamin A and 3 µg/ml of Dox to induce the SON cassette. Also, ODM + vitamin A was supplemented with 10 ng/ml of PDGF-A (PeproTech) and 10 ng/ml of IGF-1 (R&D Systems), and from day 40 onward, medium was supplemented also with 10 ng/ml of triiodothyronine (T3) (Sigma-Aldrich). The medium containing factors and Dox was changed every other day. Organoids were fixed, as previously described, at day 40 (10 d of induction) and at day 50 (20 d of induction).

## Immunocytochemistry of mixed organoids

For post-fixation, organoid slices were incubated with 4% PFA for 15 min followed by three washing steps with PBS for 5 min. HistoVT One (Thermo Fisher Scientific) antigen retrieval was performed for nuclear stainings, diluting HistoVT One 1:10 in distilled water and incubating the organoid slices in this solution at 70°C for 20 min. During the entire staining procedure, slides were kept in humidified staining chambers in the dark. Organoid slices were washed briefly with blocking solution (PBS, 4% normal donkey serum [Sigma-Aldrich], 0.25% Triton X-100 [Sigma-Aldrich]) followed by a 2-h incubation with blocking solution at room temperature. Primary antibodies were diluted in antibody solution (PBS, 4% normal donkey serum, 0.1% Triton X-100), and tissue sections were incubated overnight at 4°C. Next, after two washes using PBS for 5 min and one with PBS containing 0.5% Triton X-100 for 8 min, secondary antibodies were added diluted in antibody solution and incubated for 2 h at room temperature. Sections were washed three times with PBS for 5 min. Slides were counterstained with DAPI (Sigma-Aldrich) 1:1,000 in PBS for 5 min followed by one washing step with PBS. Lastly, organoid sections were mounted using Aqua-Poly/Mount (Polysciences). Antibodies used were selected according to the antibody validation reported by the distributing companies. Slides were stained with the following primary antibodies (1:300 dilution): chicken anti-GFP antibody (GFP-1020; Aves Labs), mouse anti-MAP2 antibody (M4403; Sigma-Aldrich), rabbit anti-Olig2 antibody (AB9610; Sigma-Aldrich), mouse anti-CNP antibody (MS-349-P; LabVision/Neomarkers), rat anti-MBP antibody (MCA409S; Bio-Rad), rabbit anti-Ki-67 (MA5-14520; Thermo Fisher Scientific), mouse PARP1 (32563; Cell Signaling Technology). The following secondary antibodies were used (1:1,000 dilution): goat anti-chicken Alexa 488 (A11039; Thermo Fisher Scientific), goat anti-mouse IgG1 Alexa 555 (A21127; Thermo Fisher Scientific), goat anti-rabbit Alexa 647 (A21245; Thermo Fisher Scientific), goat anti-rabbit Alexa Cy3 (A10520; Thermo Fisher Scientific), goat anti-mouse IgG1 Alexa 647 (A21240; Thermo Fisher Scientific), goat anti-rat Alexa 555 (A10522; Thermo Fisher Scientific).

## Microscopy and image analysis of organoid slices

Epifluorescence pictures were taken using EVOS M7000 Imaging System (Thermo Fisher Scientific). Images were analyzed using FIJI (v1.52–1.54) employing the Cell Counter plugin. The data were plotted and statistically analyzed using R (v4.4) and its packages ggplot2 (v.3.4.4) and ggpubr (v0.6.0).

## RNA preparation and quantitative RT–PCR (qRT–PCR) of 2D cultures

RNA was isolated using RNeasy Micro Kit (QIAGEN). Afterward, cDNA was generated by reverse transcription of 1 $\mu$g RNA using m-MuLV RT reverse transcriptase (New England Biolabs). For qRT–PCR, the following primers were used: *TCF4:* 5′-CAATAATGACGATGAGGACCTGAC-3′ and 5′-CTCGGACTTGCTGCTCCAG-3′, *MBP:* 5′-TTAGCTGAATTCGCGTGTGG-3′ and 5′-GAGGAAGTGAATGAGCCGGTTA-3′, *PLP1:* 5′-TGCTGATGCCAGAAT GTATGG-3′ and 5′-GCAGATGGACAGAAGGTTGGA-3′, *MOG:* 5′-TGGCAAGC TTATCAAGACCCTC-3′ and 5′-CACCTTTCCCTCACCAATAGCAT-3′, *CNP:* 5′-CG CTCTACTTCGGCTGGTTC-3′ and 5′-CCATCTTCTCCCTGGGCTCA-3′, *GALC:* 5′-TATTTCCGAGGATACGAGTGGT-3′ and 5′-CCAGTCGAAACCTTTTCCCAG-3′, *A CSS2:* 5′-TGCTTTTTACTGGGAGGGCA-3′ and 5′-TCCATGAGATCTGGGGCTGA -3′, *FASN:* 5′-GTCTTGAACTCCTTGGCGGA-3′ and 5′-GCCATCTCTCAAGACCA CGG-3′. For normalization, we used human *RPL8:* 5′-AAGGCAAA-GAGGAACTGCTG-3′ and 5′-AGCAATGAGACCCACTTTGC-3′.

Data were analyzed using the $2^{-\Delta\Delta CT}$ method as previously described (Livak & Schmittgen, 2001), and the natural logarithm thereof was used as indicated; expression levels were obtained by normalizing each sample to the endogenous *RPL8* levels.

## Immunocytochemistry and quantification of cells in 2D

For immunocytochemical analysis, cells on coverslips were fixed in 4% PFA, then permeabilized, and blocked using PBS plus 10% FCS (Anprotec), 1% BSA (Roth), plus 0.1% Triton X-100 and consecutive incubation with primary and secondary antibodies in the same solution. Afterward, DAPI staining was performed (1:10,000 in PBS), and the coverslips were embedded using Mowiol. Primary antibodies were directed against GFP (chicken, 1:500, GFP-1020; Aves Labs), NANOG (goat, 1:300; R&D Systems), OCT4 (mouse, 1:300, sc-5279; Santa Cruz), TCF4 (rabbit, 1:1,000, ab217668; Abcam), SOX2 (Y-17, goat, 1:300, sc-17320; Santa Cruz), NESTIN (mouse, 1:300, MAB5326; Merck Millipore), PAX6 (rabbit, 1:300, 901301; Biozol), cleaved caspase-3 (rabbit, 9661; Cell Signaling Technology), O4 (mouse IgM, 1:500, MAB1326; R&D Systems), MAP2 (mouse, 1:500, M4403; Sigma-Aldrich), OLIG2 (rabbit, 1:1,000, AB9610; Sigma-Aldrich), CNP1 (mouse, 1:500, MS-349-P; LabVision/Neomarkers), Ki-67 (rabbit, 1:500, MA5-14520; Thermo Fisher Scientific), MBP (rat, 1:300, MCA409S; Bio-Rad Laboratories), and SOX10 (rabbit, 1:5,000 [Stolt et al, 2003]). Alexa Fluor 488 donkey anti-goat (A11055; Invitrogen), cy3 donkey anti-rat (712-165-153; Dianova), cy3 donkey anti-mouse (711-175-150; Dianova), Cy5 donkey anti-rabbit (711-175-152; Dianova), and Cy5 goat anti-rabbit (111-175-144; Jackson) were used as secondary antibodies and all used in a 1:500 dilution. Signals were detected on a Leica DFC

350 FX microscope. Pictures were taken by an HC PL APO 20X/ 0.80 and quantified using ImageJ.

To quantify TCF4 protein levels in hiPSCs, we used multi-channel images for DAPI and TCF4. We segmented the DAPI nuclei using celldetection (v.0.4.5) (Upschulte et al, 2022) and the pretrained ginoro_CpnResNeXt101UNet-fbe875f1a3e5ce2c model and quantified the TCF4 protein levels in each segmented DAPI nucleus employing measure.regionprops of the scikit-image package (v0.21.0) in Python (v3.8.18). The integrated intensity (sum of the intensity of all pixels in a segmented nucleus) was plotted.

## Bulk RNA sequencing of NPCs

For bulk RNA sequencing, four replicates per condition of NPCs were harvested and used for RNA isolation using RNeasy Mini Kit (QIAGEN). Poly-A enrichment–based library preparation and transcriptome sequencing were performed by Novogene Europe (Cambridge, GBLibraries). Paired-end 150-bp reads were sequenced on NovaSeq 6000. Processing of the raw sequencing files was performed as described before (Frank et al, 2024). Briefly, the FASTQ files were preprocessed using fastp (v0.23.2) (Chen et al, 2018) with the following settings: qualified_quality_phred 20, unqualified_percent_limit 10, n_base_limit 2, length_required 20, low_complexity_filter enable, complexity_threshold 20, dedup enable, dup_calc_accuracy 6, overrepresentation_analysis, detect_adapter_for_pe, cut_right. The resulting FASTQ files were aligned to the human genome GRCH38 release 47 from GENCODE using R (v4.4.2) and the R package Rsubread (v2.20.0) (Liao et al, 2019). To count the aligned reads, we employed the Rsubread built-in function feature_count with the following settings: isPairedEnd = T, countReadPairs = T, require BothEndsMapped = T, countChimericFragments = T, countMulti MappingReads = F, allowMultiOverlap = F. We then used the R package EdgeR (Robinson et al, 2010) (v4.4.2) to filter genes using the EdgeR built-in function filterByExpr with default values and normalized differences in sequencing depth between samples employing calcNormFactors. PCA was performed with EdgeR and plotted using the R package ggplot2 (v.3.5.1). For the confidence ellipses, we took advantage of the R package ggpubr (v.0.6.0). To plot the expression of selected genes, the count matrix was $\log_2$-transformed with a prior.count = 2 using the EdgeR function cpm and plotted using the R package pheatmap package (v1.0.12). To calculate the activity of the regulon, we took advantage of the cross-section of empirically determined TCF4 targets in neural cells using ChIP-seq (Forrest et al, 2018; Xia et al, 2018) and functional transcriptomic data (Doostparast Torshizi et al, 2019) and used the corresponding genes in supplementary table 10 of Doostparast Torshizi et al (2019) as target genes. Genes that were down-regulated in TCF4-HOM compared with TCF4-WT in our dataset were considered to be activated by TCF4, whereas genes that were up-regulated in TCF4-HOM compared with TCF4-WT were considered to be inhibited. We next used the univariate linear model of R package decoupleR (v.2.12.0) to calculate the regulon activity. Regression analysis between TCF4 regulon activity and *TCF4* mRNA expression was done employing the R package smplot2 (v.0.2.5).

### Karyotyping

Karyotyping was performed by GTG banding at a resolution of 450–550 bands using standard protocols for lymphocyte cultures.

### Statistics and reproducibility

Data were statistically analyzed with GraphPad Prism or R using statistical tests indicated throughout the article. No statistical methods were used to predetermine the sample size. The investigators were not blinded to allocation and outcome analysis. The experiments were not randomized.

## Data Availability

The FASTQ files of the bulk RNA-seq data for this study have been deposited in the European Nucleotide Archive (ENA) at EMBL-EBI under the accession number PRJEB87238. The data that support the findings of this study are available from the corresponding authors upon reasonable request.

## Supplementary Information

## Acknowledgements

This work was supported by grants from the German Research Foundation (DFG) (KA3125/2-1; GFK2162/2 TP C2) to M Karow, the Bavarian State Ministry of Sciences, Research, and the Arts (ForInter; F.2-F2412.30/1/24) to M Karow, S Falk, M Wegner, DC Lie, and J Winkler, the German Research Foundation (DFG) project 460333672 CRC1540 EBM to M Karow and S Falk, and the Interdisciplinary Center for Clinical Research (IZKF) at the University Hospital of the FAU Erlangen-Nürnberg to M Karow (Jochen-Kalden-Funding Programme N7) and S Falk (E32).

### Author Contributions

F Furlanetto: formal analysis and investigation.
N Flegel: formal analysis.
M Kremp: formal analysis, investigation, and visualization.
C Spear: investigation.
F Fröb: formal analysis and investigation.
M Alfonsetti: investigation.
B Bohl: investigation.
M Krumbiegel: formal analysis and investigation.
S Turan: resources and investigation.
A Reis: supervision, methodology, and writing—review and editing.
DC Lie: resources and supervision.
J Winkler: resources.
S Falk: conceptualization, data curation, formal analysis, and visualization.
M Wegner: conceptualization.
M Karow: conceptualization, supervision, funding acquisition, visualization, project administration, and writing—original draft, review, and editing.

### Conflict of Interest Statement

The authors declare that they have no conflict of interest.

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
