## [Reviewer comments · Life Science Alliance]

Life Science Alliance

A novel human organoid model system reveals requirement of TCF4 for oligodendroglial differentiation

Federica Furlanetto, Nicole Flegel, Marco Kremp, Chiara Spear, Franziska Froeb, Margherita Alfonseti, Bettina Bohl, Mandy Krumbiegel, Soeren Turan, André Reis, Dieter Lie, Jürgen Winkler, Sven Falk, Michael Wegner, and Marisa Karow

DOI: <https://doi.org/10.26508/lsa.202403102>

Corresponding author(s): Marisa Karow, Friedrich-Alexander-Universität Erlangen-Nürnberg

Review Timeline:

Submission Date:	2024-10-20
Editorial Decision:	2024-12-02
Revision Received:	2025-03-13
Editorial Decision:	2025-03-14
Revision Received:	2025-03-19
Accepted:	2025-03-19

Transaction Report:

December 2, 2024

Re: Life Science Alliance manuscript #LSA-2024-03102

Prof. Marisa Karow
Friedrich-Alexander-University Nürnberg-Erlangen
Institute of Biochemistry
Fahrstrasse 17
Erlangen, Bavaria 91054
Germany

Dear Dr. Karow,

Thank you for submitting your manuscript entitled "A novel human organoid model system reveals requirement of TCF4 for oligodendroglial differentiation" to Life Science Alliance. The manuscript was assessed by expert reviewers, whose comments are appended to this letter. We invite you to submit a revised manuscript addressing the Reviewer comments.

Thank you for this interesting contribution to Life Science Alliance. We are looking forward to receiving your revised manuscript.

Sincerely,

B. MANUSCRIPT ORGANIZATION AND FORMATTING:

Reviewer #1 (Comments to the Authors (Required)):

Using SON15 hiPSC line (that is genetically modified by introducing a doxycycline (Dox)-inducible expression cassette for SOX10, OLIG2 and NKX6.2 (SON) into the AAVS1 locus) undergone CRISPR/Cas9-dependent genome editing of the TCF4 locus, the authors showed that TCF4 deficiency impaired oligodendroglial differentiation in 2D culture. Using a mixing brain organoids (a mixing approach in which we jointly aggregated control hiPSCs with SON15 hiPSCs in a ratio of 8:1), the authors showed that oligodendroglial cells could be generated in the mixing brain organoids and that heterozygous TCF4 loss impaired oligodendroglial differentiation in 3D brain organoids. The data is well written, and the data are beautifully presented. However, there are several concerns.

1. Figure 1D, E, the authors used immunochemical staining to quantify the protein level of TCF4. It could be better to quantify the protein level of TCF4 by western blot.
2. In Figure 2, the authors tried to show that TCF4 deficiency impaired oligodendroglial differentiation. To make this conclusion, it is necessary to show whether TCF4 deficiency affects OPC proliferation and apoptosis of O4+ and/or MBP+ oligodendrocytes. Moreover, does TCF4 deficiency alter the morphology of O4+ and/or MBP+ oligodendrocytes? Does TCF4 deficiency alter the protein levels of myelin proteins.
3. In Figure 3, it is important to show whether differentiated oligodendrocytes produce myelin sheaths in 3D brain organoids. If differentiated oligodendrocytes do not produce myelin sheaths in 3D brain organoids, the relevance of this model system to human physiology and pathology is very limited.
4. In Figure 2, the authors tried to show that TCF4 deficiency impaired oligodendroglial differentiation in 3D brain organoids. To make this conclusion, it is necessary to show whether TCF4 deficiency affects OPC proliferation and apoptosis of OLG2+ and/or CNP+ oligodendrocytes. Moreover, does TCF4 deficiency alter the production of myelin proteins and myelin sheaths in 3D brain organoids?

Reviewer #2 (Comments to the Authors (Required)):

In this manuscript the authors describe a novel method to generate oligodendrocytes by inducible expression of Sox10, Olig2 and Nkx6.2 (SON-transcription factors) in human iPSCs. The authors describe first differentiation into oligodendrocytes after activating the expression of these TFs in 2D. As a 3D system they mix the inducible line with control cells generating organoids to then induce expression of the SON factors, which results in oligodendrocyte generation in organoids at earlier stages and in more controlled numbers than is normally the case. The authors then proceed to use this model for disease modelling of Tcf4-deficient cells. In contrast to mouse models where only homozygous loss of Tcf4 exhibits a phenotype, using human iPSC-derived cells heterozygous for Tcf4 with the induction of the SON TFs they find clear deficits in mature oligodendrocytes, but not oligodendrocyte progenitor cells. This is certainly an interesting model that may be useful for further applications studying human oligodendrocyte diseases. However, before publication a few crucial control experiments are missing.

- 1) More quality controls for the TCF4-KO lines are needed - e.g. testing for chromosome aberrations.
- 2) It is important to examine the NSCs of controls and KOs better (bulk RNA-seq should be fine), as TCF4 is also expressed in murine and human neural stem cells (see e.g. Li et al., Mol. Psychiatry 2019).
- 3) Figure 3K: why are there many Olig2+ cells (white) that are not GFP+? This would mean not derived from the inducible cells?
- 4) This is probably asking too much, but it would be of interest to compare oligodendrocytes differentiated without forced expression of TFs to those derived from the transient SON expression. Maybe the authors could use published data and profile their oligodendrocytes?

Reviewer #3 (Comments to the Authors (Required)):

The manuscript analyzes the role of TCF4 in human oligodendroglial differentiation, a key process affected in Pitt-Hopkins Syndrome. The authors demonstrate that monoallelic and biallelic mutations in the bHLH domain of TCF4, generated through

genetic editing, significantly decrease the ability of human neural progenitor cells to differentiate and mature into myelinating oligodendrocytes. This result is based on the induced expression of SOX10, OLIG2, and NKX6.2, essential factors for oligodendroglial differentiation, and two complementary in vitro models using human iPSCs. The first is a monolayer culture with a doxycycline-activated inducible expression cassette, allowing for controlled differentiation of human oligodendrocytes. The second model employs three-dimensional cerebral organoids, where the genetically modified cells differentiate in a multicellular environment generated by unmodified cells, more faithfully reproducing the conditions of human brain tissue. The study is well-designed, the experiments are solid, and the findings are highly relevant, particularly concerning the generation of organoids containing oligodendrocytes and the investigation of TCF4 haploinsufficiency in Pitt-Hopkins Syndrome. Moreover, the proposed models represent promising tools to explore this pathology and potential therapeutic strategies, positioning the manuscript as a strong candidate for publication.

However, there are aspects that need to be improved to reinforce the robustness of the work. In Figure 2E, the data show a decreasing trend in the proportion of O4-positive cells in the HET and HOM groups compared to WT, although this difference does not reach statistical significance. This requires validation with additional replicates to determine if it reflects a genuine effect. If confirmed, it would imply that mutations in TCF4 also affect earlier stages of the oligodendroglial lineage, although to a lesser extent than maturation into myelinating oligodendrocytes. This would justify a moderate revision of the conclusions. Following this line, Figures 2G and 2H should include histograms with additional normalization to OLIG2, beyond those already performed, to mitigate biases arising from changes in the levels of this protein and consolidate the interpretation of the data on oligodendroglial maturation. Given that this normalization was used in the organoids to measure CNP (Figure 4D), the new data would enhance consistency between the models and facilitate a more precise comparison.

Furthermore, although the technical complexity of generating organoids is acknowledged and the data aim to serve as an initial basis for future studies, it would be very advisable to include additional markers of myelinating oligodendrocytes in Figure 4, specifically the marker O4, to correlate the results here with the adherent cultures. This addition would strengthen the coherence of the study and the validity of its findings. Finally, reflections on potential future developments or experiments not carried out should not be included in the results section but rather moved to the discussion section, which would improve the manuscript's structure and clarity.

Point-by-point answer to the reviewers

LSA-2024-03102

Reviewer #1

Using SON15 hiPSC line (that is genetically modified by introducing a doxycycline (Dox)-inducible expression cassette for SOX10, OLIG2 and NKX6.2 (SON) into the AAVS1 locus) undergone CRISPR/Cas9-dependent genome editing of the TCF4 locus, the authors showed that TCF4 deficiency impaired oligodendroglial differentiation in 2D culture. Using a mixing brain organoids (a mixing approach in which we jointly aggregated control hiPSCs with SON15 hiPSCs in a ratio of 8:1), the authors showed that oligodendroglial cells could be generated in the mixing brain organoids and that heterozygous TCF4 loss impaired oligodendroglial differentiation in 3D brain organoids. The data is well written, and the data are beautifully presented. However, there are several concerns.

We sincerely appreciate the reviewer's very thorough reading and evaluation of our manuscript, and for providing us with positive support, as well as constructive feedback. We have worked hard to address each major point to improve our manuscript.

1. Figure 1D, E, the authors used immunochemical staining to quantify the protein level of TCF4. It could be better to quantify the protein level of TCF4 by western blot.

Unfortunately, the commercial antibody used in our study did not work in Western Blots. Therefore, we based our conclusions about the diminished TCF4 levels on quantified immunofluorescence data. All stainings were performed simultaneously and the images are acquired with the same settings, producing pictures with pixel values only in the dynamic range of the picture, i.e. no under- or over-exposed pixel values. The quantification of the images is performed by using an automated image analysis pipeline, excluding any impact of the experimenter in the analysis. Such standardized approaches have been used to quantify protein levels before and were verified to be quantitative by mass spectrometry (Montero Llopis et al., 2021; Toki et al., 2017).

Importantly, in the new RNA-seq dataset we have furthermore taken advantage of the transcription factor function of TCF4 and assessed the expression of the TCF4 regulon in the WT as well as the TCF4-HET and TCF4-HOM samples (Figures 2C, D). Here we found a striking correlation of the *TCF4* expression levels with the TCF4 regulon activity (Figure 2E), further providing independent evidence for functional consequences of the gradual diminishment of TCF4 protein from the TCF4-HET to the TCF4-HOM samples.

2. In Figure 2, the authors tried to show that TCF4 deficiency impaired oligodendroglial differentiation. To make this conclusion, it is necessary to show whether TCF4 deficiency affects OPC proliferation and apoptosis of O4+ and/or MBP+ oligodendrocytes. Moreover, does TCF4 deficiency alter the morphology of O4+ and/or MBP+ oligodendrocytes? Does TCF4 deficiency alter the protein levels of myelin proteins.

The reviewer is right that it is very important to assess proliferation in the context of impaired oligodendroglial differentiation. In the revised manuscript we investigated proliferation and include new data showing that there is neither a significant difference in the fraction of Ki67+ SOX10+ double positive cells per SOX10+ cells, nor in the fraction of O4+ SOX10+ Ki67 triple positive cells per SOX10+ cells across experimental conditions (new Figures 2H, I).

As rightly suggested by the reviewer it is also essential to assess cell death as the underlying cause for the reduced oligodendroglial differentiation. We did in fact perform cleaved Caspase

3 stainings in the cultures, but did not detect any immunoreactive cells (TCF4-WT: 0 out of 1013 cells; TCF4-HET 0 out of 987 cells; TCF4-HOM: 0 out of 965 cells) for quantifications, therefore excluding apoptosis as underlying reason for the observed phenotype. We now mention this in the revised version of the manuscript.

We appreciate the reviewer's question on the morphology of the O4+ oligodendrocytes. In the revised manuscript we included a new analysis based on a morphological assessment of O4+ cells, where cells were either categorized as showing a more compact (less differentiated) or a more complex (more differentiated) morphology (new Figure 3F). This analysis revealed a gradual increase in the fraction of compact cells from TCF4-WT to TCF4-HET and even more to TCF4-HOM and a concomitant genotype-dependent decrease in the fraction of complex cells with the lowest fraction observed in the TCF4-HOM condition (new Figure 3G)

We also thank the reviewer for raising the important question whether the protein levels of myelin proteins are altered. We did assess the MBP intensity in the 2D cultures and found significantly lower intensities of MBP in the TCF4-HET and the TCF4-HOM samples compared to TCF4-WT. These new data were added in the new Figure 3E.

3. In figure 3, it is important to show whether differentiated oligodendrocytes produce myelin sheaths in 3D brain organoids. If differentiated oligodendrocytes do not produce myelin sheaths in 3D brain organoids, the relevance of this model system to human physiology and pathology is very limited.

We appreciate the comment of the reviewer concerning the degree of maturation of the oligodendrocytes generated within brain organoids. We want to point out the increased experimental complexity of the 3D system. For the present study, we have focused on showing that we can generate oligodendroglial cells within the 3D context of brain organoids, yet did so far not use later timepoints which would likely be needed to achieve the complex maturation of oligodendroglial cells required to produce myelin sheaths. In sum, we feel that performing electron microscopy to unambiguously show myelin sheaths in the mixed organoids is beyond the scope of this study.

4. In Figure 2, the authors tried to show that TCF4 deficiency impaired oligodendroglial differentiation in 3D brain organoids. To make this conclusion, it is necessary to show whether TCF4 deficiency affects OPC proliferation and apoptosis of OLG2+ and/or CNP+ oligodendrocytes. Moreover, does TCF4 deficiency alter the production of myelin proteins and myelin sheaths in 3D brain organoids?

The reviewer rightly asks for the assessment of proliferation and apoptosis in the absence of TCF4 in the 3D model system. In the revised manuscript we included new data addressing both proliferation as well as cell death within brain organoids. We performed co-staining of OLIG2 and KI67 and quantified the fraction of OLIG2 cells which are also KI67 positive (new Figures 5B, C). In analogy to the 2D data, we found no significant differences in SON cell proliferation between the TCF4-WT or the TCF4-HET samples.

We also performed stainings of PARP1 (Poly-(ADP-Ribose-) Polymerase 1), a target of caspase 3 in brain organoids and searched for cell death in OLIG2 cells. As shown in the new Figure (Figures 5G, H), we did not find substantial amounts of PARP1 positive OLIG2 cells in neither the TCF4-WT nor the TCF4-HET samples, thereby excluding the possibility that the reduced oligodendroglial differentiation is based on cell death of differentiating cells.

Regarding the question whether TCF4 deficiency alters the production of myelin proteins we also included another new quantification of the fraction of MBP-positive cells out of the OLIG2 positive cells which show that in the TCF4-HET condition, we find significantly less MBP positive cells (new Figure 5F).

Reviewer #2

In this manuscript the authors describe a novel method to generate oligodendrocytes by inducible expression of Sox10, Olig2 and Nkx6.2 (SON-transcription factors) in human iPSCs. The authors describe first differentiation into oligodendrocytes after activating the expression of these TFs in 2D. As a 3D system they mix the inducible line with control cells generating organoids to then induce expression of the SON factors, which results in oligodendrocyte generation in organoids at earlier stages and in more controlled numbers than is normally the case. The authors then proceed to use this model for disease modelling of Tcf4-deficient cells. In contrast to mouse models where only homozygous loss of Tcf4 exhibits a phenotype, using human iPSC-derived cells heterozygous for Tcf4 with the induction of the SON TFs they find clear deficits in mature oligodendrocytes, but not oligodendrocyte progenitor cells. This is certainly an interesting model that may be useful for further applications studying human oligodendrocyte diseases. However, before publication a few crucial control experiments are missing.

We thank the reviewer for thoroughly reading and evaluating our manuscript, and for providing great suggestions for how to improve the manuscript. We have addressed shortcomings highlighted by the reviewer as specified below.

1) More quality controls for the TCF4-KO lines are needed - e.g. testing for chromosome aberrations.

The reviewer is asking to control for chromosome aberrations in the genome edited iPSC lines to exclude that putative off targets may be the underlying cause of the observed phenotypes. We did perform karyotyping of the iPSC lines with the following results, which are now also described in the material and method section of the revised manuscript.

As mentioned in the experimental procedures of the revised version, karyotyping of the TCF4-WT cells revealed a balanced translocation between the Y chromosome and one chromosome 20. In most cells, an additional derivative chromosome 20 was present. In a small percentage of cells, an isochromosome 20q was detectable instead of the two derivative chromosomes 20. This kind of chromosomal alterations are frequent in pluripotent stem cells (Avery et al., 2013). Of note the *TCF4* gene is located on chromosome 18 and is thus not affected by the translocation. No further chromosomal aberrations were detected in either the TCF4-HET or the TCF4-HOM lines. Therefore, we assume that the detected chromosome alterations do not contribute to the observed phenotypes.

2) It is important to examine the NSCs of controls and KOs better (bulk RNA-seq should be fine), as TCF4 is also expressed in murine and human neural stem cells (see e.g. Li et al., Mol. Psychiatry 2019).

We highly appreciate the suggestion of the reviewer in including a more global assessment of the transcriptomes of the different neural stem cells. Bulk RNA-sequencing of NPCs of all conditions (TCF4-WT, TCF4-HET, TCF4-HOM) was performed and yielded very interesting insights. As now shown in the new Figures 2B-E, we can show that overall, the NPCs are not

different in regard to the loss of pluripotency markers, the presence of neuronal, astroglial or oligodendroglial markers. Furthermore, we used the bulk RNA-seq data to assess the extent of downregulation of *TCF4* expression and subsequently the impact on the TCF4 regulon activity. As shown in the new Figures 2C, D, as the mRNA expression levels of *TCF4* decrease from the TCF4-HET to the TCF4-HOM samples, also the TCF4 regulon activity decreased in a TCF4 genotype dependent manner. Here we found a direct correlation of the *TCF4* mRNA expression levels with the degree of the TCF4 regulon activity (Figure 2D), providing evidence for functional consequences of the absence of TCF4 protein in the TCF4-HET and TCF4-HOM samples.

3) Figure 3K: why are there many Olig2+ cells (white) that are not GFP+? This would mean not derived from the inducible cells?

The reviewer here raises a super interesting topic that was also discussed in the manuscript. This concerns the possibility to assess in future experiments whether the presence of OLIG2 expressing cells (derived from the induction of the SON cassette) would impact on the brain-organoid cells and induce oligodendroglial differentiation in cells without SON cassette. Yet, in the provided example this is not the case. We changed the picture in the new version of the manuscript (now relocated Figure 4K) and used different colors.

4) This is probably asking too much, but it would be of interest to compare oligodendrocytes differentiated without forced expression of TFs to those derived from the transient SON expression. Maybe the authors could use published data and profile their oligodendrocytes?

We fully agree with the reviewer that it would be interesting to assess the degree of comparability of the oligodendroglial cells generated through forced expression of the SON cassette to cells derived through directed differentiation. Yet, we feel that this is out of the scope of our study.

Reviewer #3

The manuscript analyzes the role of TCF4 in human oligodendroglial differentiation, a key process affected in Pitt-Hopkins Syndrome. The authors demonstrate that monoallelic and biallelic mutations in the bHLH domain of TCF4, generated through genetic editing, significantly decrease the ability of human neural progenitor cells to differentiate and mature into myelinating oligodendrocytes. This result is based on the induced expression of SOX10, OLIG2, and NKX6.2, essential factors for oligodendroglial differentiation, and two complementary in vitro models using human iPSCs. The first is a monolayer culture with a doxycycline-activated inducible expression cassette, allowing for controlled differentiation of human oligodendrocytes. The second model employs three-dimensional cerebral organoids, where the genetically modified cells differentiate in a multicellular environment generated by unmodified cells, more faithfully reproducing the conditions of human brain tissue. The study is well-designed, the experiments are solid, and the findings are highly relevant, particularly concerning the generation of organoids containing oligodendrocytes and the investigation of TCF4 haploinsufficiency in Pitt-Hopkins Syndrome. Moreover, the proposed models represent promising tools to explore this pathology and potential therapeutic strategies, positioning the manuscript as a strong candidate for publication. However, there are aspects that need to be improved to reinforce the robustness of the work.

We sincerely thank the reviewer for thoroughly reviewing our manuscript, as well as the kind

words, support, and productive recommendations for how to improve the manuscript.

In Figure 2E, the data show a decreasing trend in the proportion of O4-positive cells in the HET and HOM groups compared to WT, although this difference does not reach statistical significance. This requires validation with additional replicates to determine if it reflects a genuine effect. If confirmed, it would imply that mutations in TCF4 also affect earlier stages of the oligodendroglial lineage, although to a lesser extent than maturation into myelinating oligodendrocytes. This would justify a moderate revision of the conclusions.

The reviewer brought up an interesting point and we now provide new data (new Figure 3C) in which we included more data points and indeed now see statistical differences, albeit only in the TCF4-HOM condition. Furthermore, we performed a new analysis assessing the intensity of the myelin protein MBP. This revealed a TCF4 genotype dependent decrease in the MBP intensities (new Figure 3E), showing that not only the number of MBP cells decreases, but in the MBP positive cells the myelin protein is reduced.

Following this line, Figures 2G and 2H should include histograms with additional normalization to OLIG2, beyond those already performed, to mitigate biases arising from changes in the levels of this protein and consolidate the interpretation of the data on oligodendroglial maturation. Given that this normalization was used in the organoids to measure CNP (Figure 4D), the new data would enhance consistency between the models and facilitate a more precise comparison.

We thank the reviewer for this suggestion. We show in the relocated Figure 2G that the administration of Dox induced the SON cassette as shown by quantifications of SOX10 and OLIG2 in the same manner across experimental conditions. Importantly, the SON cassette contains the three factors SOX10, OLIG2, and NKX6.2 and we can use either factor as a proxy for the activity of the SON cassette upon Dox administration. Thus, we feel that using SOX10 (instead of OLIG2) for the normalization of the quantifications of O4 and MBP positive cells (new Figure 3C) does not change the conclusion.

Furthermore, although the technical complexity of generating organoids is acknowledged and the data aim to serve as an initial basis for future studies, it would be very advisable to include additional markers of myelinating oligodendrocytes in Figure 4, specifically the marker O4, to correlate the results here with the adherent cultures. This addition would strengthen the coherence of the study and the validity of its findings.

We appreciate the comment of the reviewer concerning the degree of maturation of the oligodendrocytes generated within brain organoids. We want to point out the increased experimental complexity of the 3D system. For the present study, we have focused on showing that we can generate oligodendroglial cells within the 3D context of brain organoids, yet did so far not use later timepoints which would likely be needed to achieve the complex maturation of oligodendroglial cells required to produce O4 and myelin sheaths.

Finally, reflections on potential future developments or experiments not carried out should not be included in the results section but rather moved to the discussion section, which would improve the manuscript's structure and clarity.

We thank the reviewer for the suggestion. Given the rather short nature of the manuscript we provided a combined results and discussion section, also according to the journal's policy.

References

- Avery, S., Hirst, A. J., Baker, D., Lim, C. Y., Alagaratnam, S., Skotheim, R. I., Lothe, R. A., Pera, M. F., Colman, A., Robson, P., et al.** (2013). BCL-XL mediates the strong selective advantage of a 20q11.21 amplification commonly found in human embryonic stem cell cultures. *Stem Cell Reports* **1**, 379–86.
- Montero Llopis, P., Senft, R. A., Ross-Elliott, T. J., Stephansky, R., Keeley, D. P., Koshar, P., Marqués, G., Gao, Y.-S., Carlson, B. R., Pengo, T., et al.** (2021). Best practices and tools for reporting reproducible fluorescence microscopy methods. *Nat Methods* **18**, 1463–1476.
- Toki, M. I., Cecchi, F., Hembrough, T., Syrigos, K. N. and Rimm, D. L.** (2017). Proof of the quantitative potential of immunofluorescence by mass spectrometry. *Laboratory Investigation* **97**, 329–334.

March 14, 2025

RE: Life Science Alliance Manuscript #LSA-2024-03102R

Prof. Marisa Karow
Friedrich-Alexander-Universität Erlangen-Nürnberg
Institute of Biochemistry
Fahrstrasse 17
Erlangen, Bavaria 91054
Germany

Dear Dr. Karow,

Thank you for submitting your revised manuscript entitled "A novel human organoid model system reveals requirement of TCF4 for oligodendroglial differentiation". We would be happy to publish your paper in Life Science Alliance pending final revisions necessary to meet our formatting guidelines.

- please be sure that the authorship listing and order is correct
- please add the X and Bluesky handles of your host institute/organization as well as your own or/and one of the authors in our system
- please be sure that the authorship listing and order are correct -- The full name (middle names as initials) of each author should be given.
- please rename Summary to Abstract and Experimental procedures to Materials and Methods
- the contributions selected for André Reis do not qualify them for authorship. Please either update the contributions in our system and the Author Contributions section of the manuscript or let us know if the author needs to be removed (and added eventually to the acknowledgment section)
- please consult our manuscript preparation guidelines <https://www.life-science-alliance.org/manuscript-prep> and make sure your manuscript sections are labeled correctly
- please add a callout for Figure 5F to your main manuscript text
- please remove the highlights section on the title page
- please add a Data Availability statement at the end of the Materials and Methods section indicating accession information for the RNA-seq data

A. FINAL FILES:

-- Summary blurb (enter in submission system): A short text summarizing in a single sentence the study (max. 200 characters including spaces). This text is used in conjunction with the titles of papers, hence should be informative and complementary to the title. It should describe the context and significance of the findings for a general readership; it should be written in the

present tense and refer to the work in the third person. Author names should not be mentioned.

B. MANUSCRIPT ORGANIZATION AND FORMATTING:

Sincerely,

March 19, 2025

RE: Life Science Alliance Manuscript #LSA-2024-03102RR

Prof. Marisa Karow
Friedrich-Alexander-Universität Erlangen-Nürnberg
Institute of Biochemistry
Fahrstrasse 17
Erlangen, Bavaria 91054
Germany

Dear Dr. Karow,

Thank you for submitting your Methods entitled "A novel human organoid model system reveals requirement of TCF4 for oligodendroglial differentiation". It is a pleasure to let you know that your manuscript is now accepted for publication in Life Science Alliance. Congratulations on this interesting work.

DISTRIBUTION OF MATERIALS:

Again, congratulations on a very nice paper. I hope you found the review process to be constructive and are pleased with how the manuscript was handled editorially. We look forward to future exciting submissions from your lab.

Sincerely,
